# Components of the ribosome biogenesis pathway underlie establishment of telomere length set point in Arabidopsis

Liliia R. Abdulkina [1,6], Callie Kobayashi[2,6], John T. Lovell[3], Inna B. Chastukhina [1], Behailu B. Aklilu [2], Inna A. Agabekian[1], Ana V. Suescún [2], Lia R. Valeeva[1], Chuluuntsetseg Nyamsuren[1], Galina V. Aglyamova[4], Margarita R. Sharipova [1], Dorothy E. Shippen[2]*, Thomas E. Juenger [4]* & Eugene V. Shakirov [1,4,5]*

Telomeres cap the physical ends of eukaryotic chromosomes to ensure complete DNA replication and genome stability. Heritable natural variation in telomere length exists in yeast, mice, plants and humans at birth; however, major effect loci underlying such polymorphism remain elusive. Here, we employ quantitative trait locus (QTL) mapping and transgenic manipulations to identify genes controlling telomere length set point in a multi-parent *Arabidopsis thaliana* mapping population. We detect several QTL explaining 63.7% of the total telomere length variation in the Arabidopsis MAGIC population. Loss-of-function mutants of the *NOP2A* candidate gene located inside the largest effect QTL and of two other ribosomal genes *RPL5A* and *RPL5B* establish a shorter telomere length set point than wild type. These findings indicate that evolutionarily conserved components of ribosome biogenesis and cell proliferation pathways promote telomere elongation.

[1] Institute of Fundamental Medicine and Biology, Kazan (Volga Region) Federal University, Kazan, Republic of Tatarstan, Russia 420008. [2] Department of Biochemistry and Biophysics, Texas A&M University, 2128 TAMU, College Station, TX 77843-2128, USA. [3] Genome Sequencing Center, HudsonAlpha Institute for Biotechnology, Huntsville, AL 35806, USA. [4] Department of Integrative Biology, University of Texas at Austin, Austin, TX 78712, USA. [5] Department of Biological Sciences, Marshall University, Huntington, WV 25701, USA. [6] These authors contributed equally: Liliia R. Abdulkina, Callie Kobayashi. *email: dshippen@tamu.edu; tjuenger@austin.utexas.edu; shakirov@marshall.edu

The maintenance of telomeres is a fundamental and evolutionarily conserved cellular process, with important implications for human health, premature aging, and degenerative stem cell diseases[1,2]. Species-specific telomere length homeostasis (set point) is achieved through successive organismal or cellular generations (i.e. 5–15 kb at birth in humans[3] and 1–9 kb in the flowering plant *Arabidopsis thaliana*[4]) and maintained through a balance of forces that extend (e.g. telomerase activity) and shorten (e.g. the end replication problem) the telomere tract. In humans, telomere length is likely under strong stabilizing selection, since accelerated telomere shortening is linked to age-related diseases and overly long telomeres are linked to cancer[5]. Hence, deregulation of proper telomere length homeostasis is emerging as a strong determinant of human disease.

While each species is characterized by its specific telomere length set point, different genotypes of many model organisms display substantial telomere length polymorphism[6–8]. Moreover, family and twin studies revealed that up to 80% of human telomere length variation between individuals at birth is determined genetically[9,10]. Brute force screens of gene deletion libraries in yeast have uncovered 290 genes involved in telomere length maintenance[6,11–13], but little is known about how natural sequence variation at these or other loci impact telomere length diversity. A handful of loci in humans have shown replicated association with mean telomere length, however, most of the identified SNPs explain <1% of the observed heritable genetic variation[14–17]. Consequently, identification of major causal genes remains a high priority.

Model eukaryotes have yielded valuable insight into telomere regulation. For example, a genome-wide screen for telomere length regulators in mouse identified the helicase gene *RTEL1* (ref. [18]), whose homologs were subsequently shown to control telomere length in humans and *A. thaliana*[19,20]. The experimental mapping populations in *A. thaliana* provide a valuable resource to leverage the genetic basis of genotype-specific telomere length set point and search for causal alleles[21]. Specifically, the 19-parent intercrossed and self-pollinated Multi-parent Advanced Generation Inter-Cross (MAGIC) mapping population provides an ideal resource to discover candidate loci that underlie quantitative traits. In contrast to GWAS, MAGIC lines allow detection of both rare and common variants with the power of controlled line crosses. The MAGIC population also offers the additional benefit of sampling the diversity of 19 parental genotypes, providing far more raw genetic diversity than the traditional bi-parental mapping populations[21].

Here, we utilize QTL mapping, computational candidate gene screens and transgenic manipulations to identify genes that control telomere length set point in the model plant *Arabidopsis thaliana*. We detect three QTL, with the largest on chromosome 5 explaining 42.2% of the phenotypic variation in the MAGIC population. This effect size is two orders of magnitude larger than any telomere loci previously identified in humans. Longer telomeres in this population are associated with genetic polymorphism specific to the Sf-2 genotype. Using a system genetics approach combined with T-DNA mutation knock-out analyses we uncover *NOP2A*, a ribosomal RNA methyltransferase with major roles in cellular proliferation, as a likely candidate. Loss-of-function *nop2a* mutants establish a stable and shorter set point than the wild type. While the *NOP2A* paralog *NOP2B* does not appear to regulate telomere length, loss-of-function studies of the *NOP2A* gene network implicate rRNA processing and ribosome biogenesis *RPL5A* and *RPL5B* genes as additional important regulators of telomere length. These findings establish *NOP2A* and *RPL5* genes as novel positive trans-regulators of telomere length set point in plants and implicate ribosome biogenesis and cell proliferation pathways as major regulators of telomere biology.

## Results

**QTL mapping of telomere length loci in the MAGIC population**. We evaluated the extent of natural telomere length variation among the 19 parental genotypes of the MAGIC population using the terminal restriction fragment (TRF) assay[22]. Mean telomere length (mean TRF), as measured from Southern blots with TeloTool[23], spanned 2.8 kb: the Kn-0 genotype had the shortest (2.5 kb) and the Sf-2 genotype the longest (5.3 kb) telomeres (Fig. 1, Supplementary Data 1). Broad-sense heritability among parental genotypes ($n$ replicates = 4–17) was remarkably high ($H^2 = 0.87$), confirming that telomere length set point in *Arabidopsis*, like that of other species[6,15,24], is largely heritable.

To identify genetic loci that control telomere length, we conducted quantitative trait locus (QTL) mapping on 480 MAGIC lines. As observed in the parental genotypes, mean TRF was log-normally distributed across the recombinant MAGIC population and ranged from <1.8 kb to >8 kb, suggesting a polygenic genetic architecture (polygenic/kinship $h^2 = 0.362$; Fig. 2a; Supplementary Data 2). Genetic mapping revealed a major-effect ~848 kb QTL interval on the distal end of chromosome 5 (Fig. 2b). When Sf-2 genotype probabilities at the major QTL peak were included as an additive covariate, two minor-effect QTL occurring on chromosomes 1 and the proximal end of chromosome 5 (Supplementary Fig. 1A) and a small residual LOD signal near the major effect QTL (Supplementary Fig. 1B) were observed. All three significant QTL together accounted for 63.7% of telomere length variation. When taken in isolation, allelic differences at the major-effect chromosome 5 QTL explained 42.2% of the mean TRF variation present in the MAGIC population. The minor QTL on Chromosomes 1 and 5 explained 11.5% and 10.0% of the total variance, respectively.

Allelic contributions of the 19 founder parents were normally distributed for each of the two minor QTL (Supplementary Fig. 1C), indicating that the causal variants contribute in a quantitative manner across diverse *A. thaliana* accessions. However, MAGIC lines that inherited the Sf-2 allele at the primary QTL on Chr 5 exhibited an 88.1% ($t = 9.05$, $P < 10^{-16}$, Student's two-tailed $t$-test) increase in telomere length (Fig. 2c) compared to the other 18 parental haplotypes. This indicates that a variant contributed by the Sf-2 parent is responsible for the massive difference in telomere length. Additionally, the Sf-2 parental genotype harbored the longest telomeres of all 19 MAGIC founders (Fig. 1, Supplementary Data 1). Thus, a genetic variant that distinguishes the Sf-2 parent is associated with a dramatic increase in telomere length set point.

**Screening for candidate causal genes**. While the minor QTL warrant follow-up study, we chose to focus on potential causal genes that underlie the primary QTL, both because of its large effect size, and the unique parental contribution observed therein. None of the 247 gene models within the chromosomal confidence interval of the major effect chromosome 5 QTL were previously implicated in telomere biology, suggesting that the causal DNA polymorphism is associated with a novel gene. Given the effect distribution among founder parents, we assumed that one or more Sf-2-specific variants can exist in this region, and that one or several linked causal genes may produce the observed telomere-length variation. To look for potential causal genes, we scanned the chromosome 5 QTL region in the re-sequenced genomes of all 19 parental accessions[21] for genes with known Sf-2 specific (private) DNA or gene expression polymorphisms[25]. Of the 247 genes, 110 harbored one or more Sf-2 specific DNA variants within coding or promoter (2 kb upstream of transcription start) regions (Fig. 2d, blue dots, Supplementary Data 3); 33 of these genes also displayed statistically significant differential

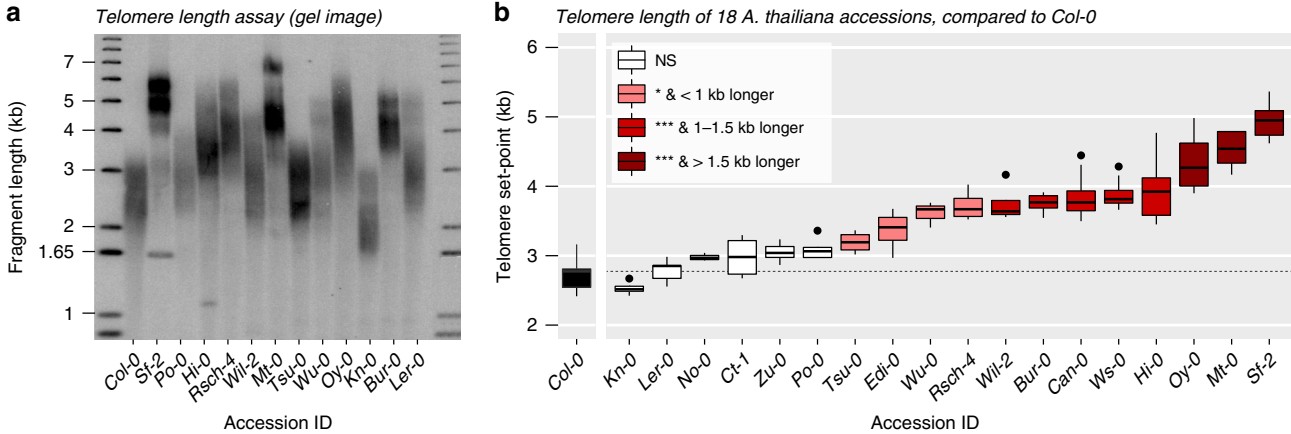

**Fig. 1** Parental genotypes of the Arabidopsis MAGIC population display different telomere length set points. **a** A representative TRF southern blot of genomic DNA from Arabidopsis accessions (genotypes). **b** Between 4 and 17 individual plants were analyzed for each accession. The mean telomere length distribution of each genotype is shown in boxplots, where the 'boxes' represent the interquartile range (IQR), the whiskers extend to the largest and smallest observations within 1.5 * IQR and points represent observations outside of the 1.5 * IQR. The boxes are colored by the significance of a Tukey's HSD test between each parent and the reference genotype, Col-0. Significance codes: NS: $P > 0.05$, *$P \geq 0.005$, ***$P < 0.005$.

expression between Sf-2 and all other parental MAGIC accessions (Supplementary Data 4). To further rank candidate genes, we examined annotations of the 110 genes with Sf-2 specific polymorphisms and its subset of 33 genes with expression differences, specifically focusing on loci related to DNA/RNA binding or metabolism, cell proliferation or chromosome biology. Overall, 20 candidate genes with Sf-2 specific polymorphisms that either showed a unique expression pattern in Sf-2 or interesting annotations or both were selected for further analysis (Fig. 2d, orange dots).

TRF analyses of T-DNA insertion lines ($n$ replicates = 2–8) revealed that mutants of 19 out of 20 candidate genes harbored telomeres in the wild type range of the Columbia (Col-0) accession (Fig. 2d, gray shaded area, Supplementary Data 5). However, telomeres in the SALK_129648 (*nop2a-2/oli2-2*) mutant, which harbors a homozygous T-DNA insertion within the *NOP2A/OLI2* (*OLIGOCELLULA 2*, At5g55920) gene, were 27.17% shorter than wild type (Fig. 2d). *NOP2A/OLI2* encodes a homolog of human NOP2, a putative S-adenosyl-L-methionine-dependent methyltransferase superfamily protein[26]. *NOP2* proteins are highly conserved[26] and participate in ribosome biogenesis, cell proliferation, and cancer progression[27,28]. The *Arabidopsis NOP2A/OLI2* gene is likewise implicated in ribosome biogenesis and control of organ size by promoting cell proliferation[29,30].

**NOP2A is a positive trans-regulator of telomere length.** To confirm the role of *NOP2A* in telomere length regulation, we analyzed an additional and independent homozygous T-DNA mutant of *NOP2A*, *nop2a-3* (SALK_082871) (Fig. 3a). Both *nop2a-2* and *nop2a-3* exhibit shorter telomeres (Fig. 3b, c). Furthermore, we measured telomere length on five individual chromosome arms 1 L, 3 L, 4 R, 5 L, and 5 R that harbor unique subtelomere sequences. Telomeres of all tested chromosome arms were shortened in both mutants, demonstrating that *NOP2A*-driven telomere shortening is global (*trans*-acting) and not specific to any one chromosome arm (Supplementary Fig. 2).

Analysis of the published 1001 Arabidopsis genomes[31] indicates that many SNPs are present in the 18 MAGIC founder genomes over Col-0 reference throughout the *NOP2A* gene (Supplementary Data 6), though Sf-2 specific SNPs are localized only in the 3′ and 5′ regulatory regions. Given that nucleotide changes in regulatory regions often lead to gene expression

differences, we tested if *NOP2A* gene expression change in Sf-2 correlates with telomere length. Of the 480 MAGIC lines used in our study, only 8 lines harboring Sf-2 haplotype and 201 lines with alternative haplotypes at the major Chr 5 QTL have been previously assayed for gene expression[32]. Comparison between lines with Sf-2 *NOP2A* haplotypes and all others revealed marginally significantly elevated *NOP2A* expression in Sf-2-harboring MAGIC lines ($t = 2.35$, df = 9.68, $P = 0.041$) (Supplementary Fig. 3A). We also experimentally assayed *NOP2A* gene expression in all 19 founder genotypes by qPCR (Supplementary Data 7) and plotted expression levels against measured mean telomere length in these genotypes (Supplementary Fig. 3B). Similarly, we plotted mean TRF - *NOP2A* expression correlations across the MAGIC lines that were subjected to gene expression assays (Supplementary Fig. 3C). Altogether, these experiments confirmed that *NOP2A* expression in Sf-2 and Col-0 differs by over 2-fold, but also indicate that *NOP2A* gene expression alone cannot fully explain observed telomere length differences in the 19 founders.

We next used genetic complementation as an additional test for causality for the observed QTL effect, where *nop2a-2* mutants (Col-0 background) were transformed with wild type *AtNOP2A* alleles from either Col-0 or Sf-2 accessions. In primary transformants T1, both *NOP2A* allelic transgenes elongated *nop2a-2* mutant telomeres up to the wild type Col-0 length (Supplementary Fig. 4). However, no significant difference in complementation efficiency was observed between Col-0 and Sf-2 alleles in this experimental setting, indicating that *NOP2A* causality for the chromosome 5 QTL cannot be unambiguously established from this experiment. One possible explanation for similar complementation by Col-0 and Sf-2 alleles in the T1 generation is that the set point length of the Sf-2 allele may require multiple generations to accrue, as is often observed in crosses between genotypes with different telomere length[8]. Nevertheless, these complementation data demonstrate that *NOP2A* alleles from both Col-0 and Sf-2 are functional and, together with analysis of multiple loss-of-function *nop2a* mutants, define *NOP2A* as a critical positive trans-regulator of telomere length in *A. thaliana*.

**NOP2A mutants establish shorter telomere length set point.** Telomere integrity requires that proper genotype-specific telomere length is established and stably maintained before and after

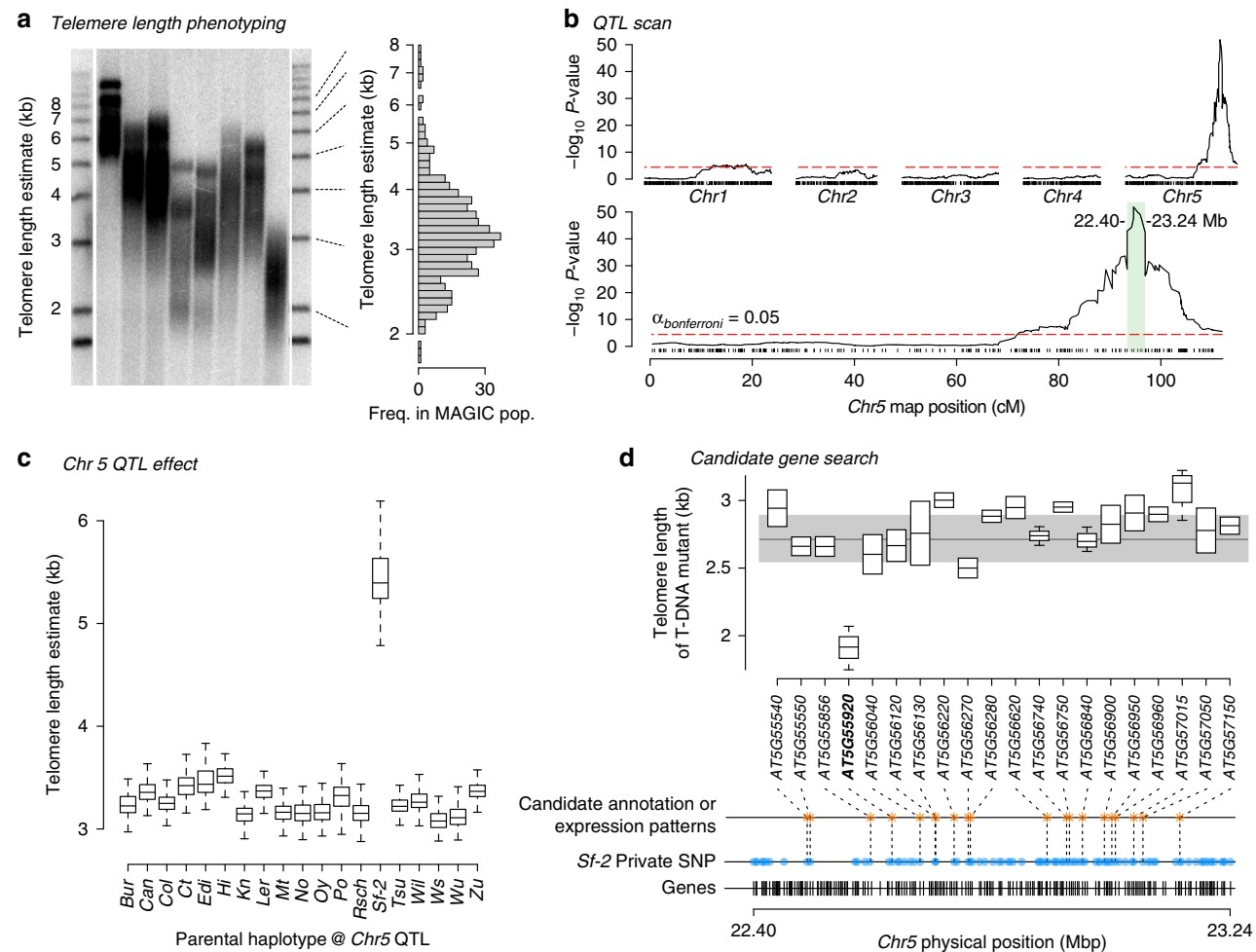

**Fig. 2** Genetic mapping of telomere length variation in Arabidopsis. **a** Representative TRF gel of telomere length variation in *A. thaliana* MAGIC lines. Each lane corresponds to genomic DNA from a randomly selected MAGIC line. Overall pattern of mean telomere length distribution (in kb) in the MAGIC lines is shown on the right. **b** QTL scan of mean telomere length in the MAGIC population. The majority of natural genetic variation in telomere length of Arabidopsis MAGIC lines is explained by a large effect QTL on chromosome 5 (upper panel). The horizontal lines indicate the genome-wide Bonferroni-corrected threshold for QTL significance (alpha = 0.05). The major effect QTL is localized to ~848 kb on the right arm of chromosome 5 (lower panel). **c** Estimates of contribution of 19 parental haplotypes at the major effect chromosome 5 QTL to telomere length (in kb) in *Arabidopsis thaliana* MAGIC lines. Plants that inherited the rare Sf-2 allele show a 88.1% increase in telomere length. Error bars represent standard errors for each haplotype. **d** The high confidence ~848 kb QTL interval on chromosome 5 contains 247 genes (black bars, bottom), of which 110 genes harbor one or more Sf-2 specific polymorphisms (blue dots). Of these, 20 genes were selected for transgenic analysis (orange asterisks) due to the presence of private Sf-2 alleles, statistical differences in expression between Sf-2 and other parental accessions or due to interesting annotation. Knockouts of 19 of these genes harbor telomeres in the normal range for the wild type Col-0 accession (gray shaded area). Transgenic T-DNA knockouts of At5g55920 gene (in bold) showed the strongest effect on telomere length, indicating that variation at this locus may be responsible for the observed QTL.

chromosomal replication. Mutations that compromise telomere maintenance lead to progressive shortening of telomeres across generations, ultimately triggering genome instability and cell cycle arrest[33]. Conversely, mutations that alter length set point cause telomeres to gradually lengthen or shorten until a new length equilibrium is established. To test whether mutation of *NOP2A* impacts telomere maintenance or set point, we followed telomere dynamics through parent-progeny analysis in *nop2a-2* and *nop2a-3* mutants. These mutants were obtained from the seed stock center in the homozygous mutant state and were likely propagated for several generations in the absence of *NOP2A* prior to our analysis. Consistent with establishment of a new shorter set point, telomere length remained unchanged in three consecutive generations of *nop2a-2* and *nop2a-3* mutants (Fig. 3c and Supplementary Fig. 5A, B).

To unambiguously test the trans-generational effect of loss-of-function *nop2a* alleles, we examined telomeres in the heterozygous *nop2a-4*$^{+/-}$ T-DNA line (Fig. 3a) and its homozygous *nop2a-4*$^{-/-}$ mutant and wild type *nop2a-4*$^{+/+}$ self-pollinated progeny. Heterozygous *nop2a-4*$^{+/-}$ plants exhibited a wild type telomere profile (Fig. 3d), indicating that *NOP2A*, like other *A. thaliana* telomere maintenance genes[22,34], is not haploinsufficient. In contrast, telomeres in the first three homozygous *nop2a-4*$^{-/-}$ mutant generations (G1–G3) shortened progressively (Fig. 3d, e), but upon reaching a mean length of 1.8–2.0 kb in G3, became stabilized and were maintained at this set point in G4 and G5 generations (Fig. 3e, Supplementary Fig. 3C). Thus, plants deficient for *NOP2A* establish a new stable telomere length equilibrium within three generations that is 27% shorter than in wild type Col-0.

**Ribosome biogenesis genes underlie telomere length control.** Arabidopsis *OLI2/NOP2A* was previously identified in a forward

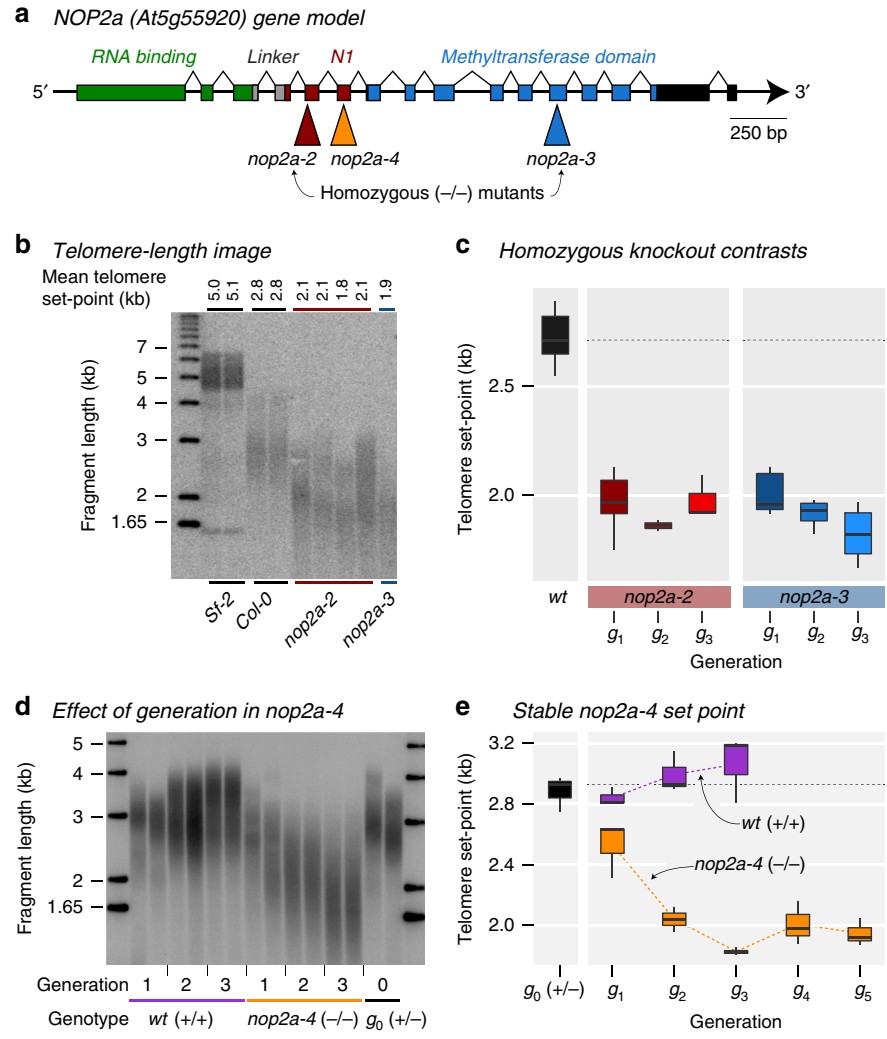

**Fig. 3** Inactivation of *AtNOP2A* gene leads to the establishment of a new shorter telomere length set point. **a** Representation of the Arabidopsis *NOP2A/OLI2* gene model. Colored boxes represent exons; black lines, introns; and black boxes, untranslated regions. Positions of T-DNA insertion sites and the corresponding mutant line numbers are indicated. Predicted S-adenosyl-ʟ-methionine-dependent methyltransferase functional domains are indicated on top. **b** A representative TRF southern blot for Sf-2 and Col-0 wild types and two *NOP2A/OLI2* homozygous T-DNA mutant lines (*nop2a-2* and *nop2a-3*) is shown, where mean telomere length (mean TRF) for each individual analyzed plant is indicated at the top. Molecular weight DNA markers (in kb) are shown on the left. **c** The distributions of telomere length in ≥3 biological replicates of the Col-0 wild type and the two homozygous mutants are shown in boxplots (midline = median, box = interquartile range (IQR), whiskers = 1.5*IQR). Across generations, T-DNA mutants show shorter telomere length set point than the corresponding wild type Col-0 genotype. **d** An example of a TRF southern blot for telomere length analysis of several consecutive generations of homozygous *nop2a-4*$^{-/-}$ mutants. Seeds of a single self-pollinated heterozygous *nop2a-4*$^{+/-}$ plant were grown to identify wild type, heterozygous and homozygous G1 progeny, which were then used to generate corresponding G2 and G3 mutant generations by single seed descent. **e**, T-DNA mutants with the null *nop2a-4* allele displayed significantly shorter telomeres after only a single generation of self-pollination, but established a new stable set point by the third generation of mutants.

genetic screen for *OLIGOCELLULA* cell proliferation mutants together with two other genes involved in rRNA binding and ribosome biogenesis, *OLI5/RPL5A* and *OLI7/RPL5B*[29,30]. Arabidopsis interactome data indicate that NOP2A physically interacts with RPL5A and RPL5B proteins (Fig. 4a). *RPL5A* and *RPL5B* encode highly conserved variants of ribosomal protein 5, an essential structural component of the large 60 S ribosomal subunit[35]. Human NOP2 and RPL5 proteins also interact[36]. Human *RPL5* inhibits tumorigenesis, and its inactivation is the most common (11–34%) somatic ribosomal protein defect in multiple tumor types[37,38]. Similar to our analysis of Arabidopsis *NOP2A/OLI2* mutants, plants lacking *RPL5A/OLI5* and *RPL5B/OLI7* display a shorter telomere length set point that is within (*rpl5a/oli5-2*) or close to (*rpl5b/oli7-2*) the 1.8–2.0 kb length established in *nop2a* mutants (Fig. 4b–d). Thus, all of the

ribosomal proteins identified in the *OLIGOCELLULA* cell proliferation pathway (NOP2A/OLI2, RPL5A/OLI5 and RPL5B/OLI7) control telomere length set point in Arabidopsis.

*NOP2A* belongs to a highly conserved[26] RCMT2 subfamily of eukaryotic RNA (cytosine-5)-methyltransferases, with most organisms having a single *NOP2* gene[39]. In contrast, *A. thaliana* offers an exceptional system to study the effects of *NOP2* gene duplication, as it harbors two *NOP2* paralogs, *NOP2A* and *NOP2B*. These two genes appear to be functionally redundant in ribosome biogenesis as double *nop2a nop2b* mutants are never recovered[40]. The availability of two *NOP2* copies in Arabidopsis afforded us an opportunity to investigate if both NOP2 proteins contribute to telomere length control or if this function is specific to NOP2A. Notably, telomere length was unaltered in two different *NOP2B* T-DNA mutants, *nop2b-1* and *nop2b-3* (Fig. 4e,

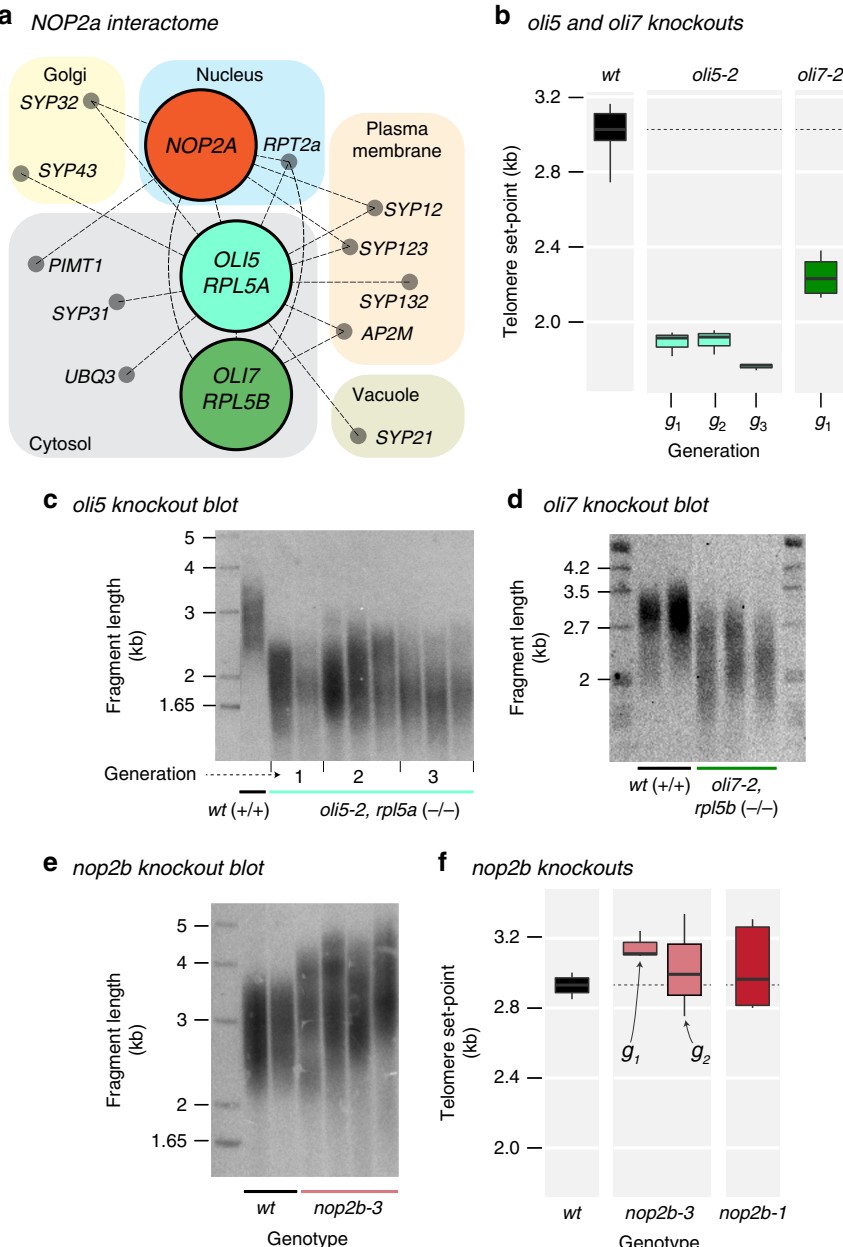

**Fig. 4** Specific components of ribosome assembly and rRNA biogenesis pathway control the establishment of telomere length set point in Arabidopsis. **a** Protein interactions ('interactome') of OLI2/NOP2A, OLI5/RPL5A, and OLI7/RPL5B, genes previously identified in a forward genetic screen for *OLIGOCELLULA* cell proliferation mutants[29], are shown. The edges in the network connect predicted interacting proteins (nodes), which are grouped by their predicted cellular locations. **b** Boxplots indicate the significantly shorter telomeres in *rpl5a/oli5-2* and *rpl5b/oli7-2* mutants relative to the wild type. **c**, **d** Examples of TRF southern blots of *rpl5a/oli5-2* and *rpl5b/oli7-2* mutants. Arabidopsis *RPL5A/OLI5* (**c**) and *RPL5B/OLI7* (**d**) T-DNA knockout plants harbor stable telomere phenotypes with a new shorter length set point. **e** Example of TRF southern blot for *nop2b-3* mutants. Unlike *NOP2A* mutants, Arabidopsis *NOP2B* T-DNA knockout plants harbor telomere length in the wild type range. **f** Boxplots (midline = median, box = interquartile range (IQR), whiskers = 1.5*IQR) indicate telomere lengths in *nop2b-1* and *nop2b-3* mutants relative to the wild type. The y-axis scale has been extended to facilitate comparisons with the minimum TRF of *rpl5a/oli5 and rpl5b/oli7* in **b**.

f), arguing that in addition to the roles of *NOP2A* and *NOP2B* in ribosome biogenesis, *NOP2A* also has a unique and separate function in telomere biology.

## Discussion

Previous genetic screens in yeast identified ribosome biogenesis as one of the largest gene categories linked to telomere length[6,11–13]. Furthermore, several human rRNA maturation proteins (i.e. Dyskerin and NOP10) interact with the telomerase RNA subunit and are implicated in the telomeropathy dyskeratosis congenita[41,42]. Indeed, the pleiotropic effects of mutations in *Dyskerin*, *NOP10* and *TERT* may account for some similarities between telomeropathies and ribosomopathies[43]. These observations, combined with our findings in *Arabidopsis*, argue that components of rRNA maturation machinery may impact species-specific telomere length set point across eukaryotic evolution.

A physical association between human NOP2 and the catalytic telomerase subunit hTERT was recently shown to stimulate a non-telomere function of telomerase, transcription of G1 phase cyclin D1 (ref. [44]). While the question of whether NOP2

modulates telomere length in humans needs to be experimentally tested, our data in Arabidopsis indicate that NOP2A has a critical role in telomere length homeostasis. Moreover, the NOP2-telomerase interaction reveals a potential mechanism of how Arabidopsis NOP2A could simultaneously influence rRNA biogenesis, cell proliferation, and telomere length homeostasis.

## Methods

**Plant materials.** Seeds for the set of 480 MAGIC recombinant inbred lines (CS782242) were obtained from Dr. Paula Kover, University of Bath. MAGIC founder accessions Bur-0 (CS6643), Can-0 (CS6660), Col-0 (CS6673), Ct-1 (CS6674), Edi-0 (CS6688), Hi-0 (CS6736), Kn-0 (CS6762), Ler-0 (CS20), Mt-0 (CS1380), No-0 (CS6805), Oy-0 (CS6824), Po-0 (CS6839), Rsch-4 (CS6850), Sf-2 (CS6857), Tsu-0 (CS6874), Wil-2 (CS6889), Ws-0 (CS6891), Wu-0 (CS6897), and Zu-0 (CS6902) were purchased from ABRC. Arabidopsis mutant lines tert[22], nop2a-2/oli2-2 (refs [29,40]) (SALK_129648), rpl5a/oli5-2/ae6-2 (refs [29,45]) (SALK_089798c), nop2b-1 (ref. [40]) (SALK_084427), and rpl5b-2/oli7-2 (refs [45,46]) (SALK_010121) were purchased from ABRC. Other mutant lines described here are nop2a-3/oli2-3 (SALK_082871, genotyped with primers SALK-55920-LP, TTGTCCAAAGATGGCTGAAAC and SALK-55920-RP, AGGCTTAAGTCGC-TAACTGCC), nop2a-4/oli2-4 (SAIL_1279_H03, genotyped with primers SAIL-55920-LP, CATTTGTCCAGGAGCTTGAAG and SAIL-55920-RP, AGAATTG GAATCATCACACGC), and nop2b-3 (SALK_117497, genotyped with primers 26600-LP2, CTAAAATTATGGGGCTGGAGG and 26600-RP2, GAGACAAGC ACGAGAGGAATG).

**Plant growth and transformation.** Wild type *Arabidopsis thaliana* seeds were cold-treated overnight at 4 °C, placed in an environmental growth chamber and grown under a 16-h light/8-h dark photoperiod at 22 °C. Seeds of T-DNA mutants were vernalized at 4 °C for 2 days, sowed on 1/2 MS medium and grown in a culture room at 22 °C with 16 h light/8 h dark photo period. Ten-day-old seedlings were transplanted to soil (3:1 ratio of Pro-Mix Bio-fungicide and Profile Field and Fairway Calcined Clay) and grown in a plant growth chamber under the same conditions.

For complementation experiments, *NOP2A* locus with its putative native promoter and regulatory sequences (1062 bp upstream of translation start codon ATG and 800 bp downstream of translation stop codon) from Sf-2 and Col-0 accessions was cloned into pCBK05 destination vector carrying the *bar* gene as a selectable marker using SbfI and BamHI restriction enzymes (pCBK05:NOP2A). Complementation constructs were introduced into the *Agrobacterium tumefaciens* GV3101 strain, which was used to transform homozygous nop2a-2/oli2-2 plants (SALK_129648 line). T1 primary transformants were selected on 1/2 MS basal medium supplemented with 25 mg/l of phosphinothricine (BASTA) (Crescent Chemical, Islandia, NY), genotyped by PCR to identify plants harboring the *NOP2A* transgene from Sf-2 or Col-0 accessions and analyzed by RT-PCR to verify transgene expression.

**Telomere length analysis.** Genomic DNA from individual whole plants was extracted and telomere length (TRF) analysis was performed with genomic DNA digested with Tru1I (Fermentas, Hanover, MD) restriction enzyme. $^{32}$P 5′-end-labeled $(T_3AG_3)_4$ oligonucleotide was used as a probe[22]. Radioactive signals were scanned by a Pharos FX Plus Molecular Imager (Bio-Rad Laboratories), and the data were analyzed by Quantity One v.4.6.5 software (Bio-Rad). Mean telomere length (mean TRF) was calculated using the TeloTool program[23]. Telomere PETRA was performed using primers to 1L, 3L, 4R, 5L, and 5R chromosome arms[47]. Source data for gels are provided in a separate Source Data file.

**Multi-parent QTL mapping.** We conducted QTL mapping in the *A. thaliana* multi-parent advanced generation intercross (MAGIC) population[21] using mean telomere length data from 480 MAGIC lines (n replicates = 1–2). All measurements were taken from distinct samples. The genotype matrix was imputed using shallow resequencing data[25] via the construct.haplotype function within the HAPPY software package[21]. We employed the accompanying happy.hbrem algorithm available in the R environment for statistical computing to infer QTL peaks. Confidence intervals were constructed around QTL peaks using a –log₁₀ P-value 'drop' of 7.5 (ANOVA). That is, we assumed the causal locus of a QTL peak existed within an interval where the P-value was no more than 7.5-orders of magnitude greater than the peak minimum P-value. This provided a very conservative genomic region with which to screen candidate genes.

In one-way scans (like we implemented above), large effect QTL can introduce large residual variance at other genomic locations. Therefore, it is often necessary to control for the effects of large QTL by conducting one-way scans conditioned on the presence of the primary QTL. Since HAPPY does not permit nuanced multiple QTL modeling, we examined the significance of minor QTL in R/qtl2 (ref.[48]), by including the probabilities of the Sf-2 haplotype at the primary QTL as an additive covariate. The proportion of phenotypic variance explained (PVE) is calculated by transforming the conditional LOD score at each peak where $L = LOD$ score and $N$ = number of individuals, then $PVE = 1 - 10 \wedge (- (2 / N) * L)$. Normalized

(FPKM) expression values of *NOP2A* were downloaded (GEO GSE94107) and used for *NOP2A* expression analysis in MAGIC lines.

**Computational screen for candidate genes.** We screened for candidate genes within the QTL interval on chromosome 5 using a combination of gene expression and DNA sequence polymorphism. Allelic effect distributions at the QTL peak demonstrated that the Sf-2 parent contributed a private allele that was responsible for causing telomere length variation. To infer candidate genes, we counted all SNPs that were private to the Sf-2 parental genome for each gene in the interval. We also compared parental gene expression[25] via a t-test contrasting voom-normalized expression[49] of the Sf-2 parent to the other parental expression values. Genes with ≥1 Sf-2 private SNP in the CDS and FDR-corrected P-value (moderated two-tailed statistics) of differential expression ≤0.05 were considered candidates. We dropped several genes from this list that were unlikely to be related to telomere-length variation (e.g. chloroplast import apparatus). Genes within the QTL region with promising annotated functions (e.g. DNA topoisomerase, transcription factors) were also included in the analysis. To visualize NOP2A protein interaction network, the Arabidopsis interaction viewer v2.0 (http://bar.utoronto.ca/interactions2/) was used.

**Quantitative RT-PCR.** Total RNA was extracted from 0.2 g of healthy plant tissue using Direct-zol RNA MicroPrep with on-column DNase I digestion for 1 h (ZYMO RESEARCH). In all, 1 µg of purified RNA was combined with random primers for cDNA synthesis using SuperScript III Reverse Transcriptase (Invitrogen) following manufacturer instructions. For quantitative RT-PCR (qPCR), 2 µl of 1:10 diluted cDNA was used in a 20 µl reaction containing 10 µl SsoAdvanced Universal SYBR Green Supermix (Bio-Rad) and 0.5 µM final concentration of each primer (Oli2_Fwd 5′-GAGCGGATTGTGGATGTAGCTG-3′ and Oli2_Rev 5′-A CGATGCAGATTGGCAGTTAGC-3′). Reactions were run on a CFX Connect Real-Time System (Bio-Rad) using 58 °C primer annealing with 15 s extension. mRNA levels were normalized to GAPDH mRNA levels in corresponding samples. For each genotype, RNA from 3 to 6 biological replicates (individual plants) were analyzed, and three technical replicates were run for each sample. Data is reported as the mean of the replicates and the error bars indicate the standard error of the mean (SEM) of biological replicates. The P-values were calculated with the Student's t-test (**<0.01; *<0.05).

**Reporting summary.** Further information on research design is available in the Nature Research Reporting Summary linked to this article.

## Data availability

Source data for Figs. 1a, 2a, 3b, d, and 4e and Supplementary Figs. 2C and 5B, C are provided in a Source Data file. Additional supporting data are provided as Supplementary Data 1–7. Details are provided in the Description of Additional Supplementary Files and the Reporting Summary.

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

## Acknowledgements

We thank Dr. Paula Kover (Milner Centre for Evolution, University of Bath) for sharing MAGIC seeds, and members of our labs for insightful discussions. This work was supported by National Institutes of Health (R01 GM065383 to D.E.S; R01 GM127402 and R03 AG052891 to E.V.S.); National Science Foundation (IOS-0922457 to T.E.J.), Russian Foundation for Basic Research (18-34-00629 to L.R.A.) and funds from the Program of Competitive Growth of Kazan Federal University. J.T.L. was supported by a National Science Foundation IOS fellowship (IOS-1402393).

## Author contributions

All authors contributed significantly to this work. T.E.J., D.E.S., M.R.S. and E.V.S. designed the experiments. L.R.A., L.R.V., C.N. and E.V.S. analyzed telomere length in MAGIC lines and parental accessions. J.T.L. performed QTL analysis. I.B.C., C.K., B.B.A., G.V.A., I.A.A. and E.V.S. analyzed T-DNA mutants and complementation lines. A.V.S performed qPCR. C.K., A.V.S. and E.V.S. performed quantitative complementation. J.T.L., D.E.S., T.E.J., and E.V.S. wrote the paper with contributions from all other authors.

## Competing interests

The authors declare no competing interests.
