## [Peer Review File · Nature Communications]

Reviewers' comments:

Reviewer #1 (Remarks to the Author):

The study uses 480 arabidopsis MAGIC lines to map a major QTL that accounts for about half of the variance for telomere length in the MAGIC population. Using a combination of haplotype analysis and gene expression in the 19 founders the authors show that an allele private to one of the 19 founders, namely Sf-2, drives the QTL and is likely to be the gene NOP2A. Using T-DNA insertion lines they confirm that knocking out this gene does indeed reduce telomere length.

This is a nice study. I am not an expert on the molecular biology of telomere length so can only really comment on the statistical genetic aspects, which appear sound as far as I can tell from the very condensed Nature Comms style.

I have a few questions, comments and suggestions.

1. How many biological replicates were phenotyped for the founders, the MAGIC lines and the T-DNA lines? I assume there were biological replicates in order to estimate broad-sense heritability.

2. In the abstract it is stated that more than one QTL is mapped but I could find very little discussion of this point. If one performs a conditional analysis, either by removing those MAGIC lines (~1/19 or 4.5%) that carry Sf-2 at the major QTL, or by including the QTL haplotype as a covariate, do further QTL appear? In particular, can one rule out more than one causal variant contributing to the major QTL? Removing the major QTL would reduce the residual variance by about half and therefore could help clarify the significance of the QTLs elsewhere which just exceed the significance threshold.

3. How common is the Sf-2 haplotype in the 1001 Arabidopsis genome survey? Is there any evidence of geographical origin of the haplotype?

4. More generally, what is the biological importance of this finding? Judging from figure 3, hets for the T-DNA insertions are recessive. However, the Sf-2 allele appears to increase telomere length, ie it is like a gain of function allele. Do the annotated Sf-2 sequence variations in NOP2A suggest a change to gene function (ie a changed protein sequence), or is the effect simply due to a change (presumably increase) in expression?

5. I assume the founder gene expression data used in the study is from ref 42 (Gan et al) - please clarify. What is the pattern of expression of NOP2A among the founders? There is also MAGIC line expression data available from Imprialou et al 2017 Genetics (available from GEO under series number GSE94107). These data would be useful to (a) confirm the founder expression patterns, (b) to see if the gene interaction module in Fig 4A is supported by MAGIC gene co-expression, (c) identify any other genes whose expression is correlated with NOP2A.

6. Please clarify if there is a distinction between telomere set point and telomere length.

Referee: Richard Mott

Reviewer #2 (Remarks to the Author):

Abdulkina et al. describe the role of NOP2A, an RNA methyltransferase involved in ribosome biogenesis, in telomere length set point establishment. For this, they exploit a genetic approach aiming at identifying the causal factor for natural variation in telomere length in Arabidopsis

thaliana. Although they convincingly demonstrate that NOP2A, and other players in ribosome biogenesis, is involved in regulating telomere length set point, I am less convinced that allelic variation for this gene explains the natural variation observed.

For instance, the broad sense heritability among parental genotypes is very high and the range of variation is quite wide, with evenly distributed phenotypic values along the spectrum (Fig. 1). Yet, a QTL analysis only detects a single relevant QTL which only seems to explain the extreme phenotype of accession Sf-2 (Fig. 2). So, given the high heritability, what does explain the large variation in telomere length in the other accessions and why could this not be mapped?

Is it possible that the true causal gene was not selected for KO screening?

The phenotype suggests a gain of function in Sf-2 compared to all other accessions. This is hard to proof with a KO or sequence analysis. If the causal gene is a repressor, however, a KO would mimic the Sf-2 phenotype and loss of function mutations are much better predictable and more common. I have no doubt that NOP2A plays a role in telomere length but question whether this gene is linked to the observed variation.

Furthermore, they test a KO of NOP2A in the Col background, which is one of the accessions with the shortest telomeres already, to show that a null-mutant results in even shorter telomeres, while the Sf-2 allele should result in longer telomeres. Perhaps a Crispr/Cas KO in the Sf-2 background provides more insight on the causality of the NOP2A locus. It would also be good to know the sequence variation between Col and Sf-2 and its anticipated effect.

A complementation test might also provide evidence for allelic variation at the NOP2A locus explaining the variation between Col and Sf-2. However, this test seems to do the opposite. Allelic variation in the Fs-2 gene does not lead to a difference in phenotype and complementation does not result in Fs-2 like trait values (carefully hidden in the extended data figure 2), making it unlikely that NOP2A is the causal gene underlying the QTL.

So, while a carefully conducted genetic screen identifies a strong QTL explaining variation in telomere length, it turns out that this QTL only explains (perhaps only partly) the extreme phenotype of accession Sf-2. They further provide compelling evidence that NOP2A and related genes are involved in the regulation of telomere length. However, they fail to prove that NOP2A is actually the gene underlying the detected QTL, nor do they provide additional loci that might explain the highly heritable variation between the other accessions.

Finally, the authors repeatedly hint at the evolutionary conserved role of NOP2 and speculate that this gene might also determine telomere length in humans and other species. I would argue that when this aspect is emphasized so strongly some empirical evidence supporting this theory could be expected.

Reviewer #3 (Remarks to the Author):

The manuscript by Shakirov and colleagues uses a quantitative genetics approach where 19 distinct isolates of Arabidopsis were crossed together to create hundreds of recombinant inbred lines whose telomere length was quantified by Southern blotting. The authors defined a locus on chromosome 5 that was present in the strain with the longest telomeres of the 19 telomeres, Sf2, which accounted for about 50% of the variation in telomere length (the locus of largest effect). Of note, the parental Sf2 genotype had a telomere length of about 5 kb, whereas the recombinants had telomere lengths up to 8 kb. The authors defined ~150 genes in the Sf2 interval that controls telomere length, 110 of which harbored Sf2-specific DNA variants that might affect gene expression or function. The authors further studied 20 candidate genes in this interval and found that independent mutations in the NOP2 gene induced telomere shortening, based on analysis of

bulk telomere length and of independent telomeres, which allays any concern that the location of NOP2 near a telomere of chromosome 5 might be responsible for the effect of the NOP2 interval on telomere length. The NOP2 gene is known to promote ribosome biogenesis, deficiency for which causes aplastic anemia in humans. Loss of telomerase also causes aplastic anemia in humans, but the mechanism by which ribosome proteins deficiency induces senescence of blood cells is not clear. The authors elegantly complement the short telomere phenotype of NOP2-deficiency strains with NOP2. The authors demonstrate that the effect of NOP2 mutation on telomere length is immediate and not progressive, in contrast to deficiency for telomerase. This implies that NOP2 might not affect telomerase or the telomerase RNA directly. Moreover, the authors show that NOP2 is not haploinsufficient. NOP2 has been previously identified based on genetic and phenotypic analysis to promote cell proliferation with two additional genes RPL5A and B. Mutations in or near both of these genes induced telomere attrition. This establishes a clear relationship between several novel ribosome RNA biogenesis factors and telomere length. Another ribosome biogenesis factor, Dyskerin, was the first gene defined to regulate telomerase in humans, although in the case of human dyskerin deficiency, the cause of disease is shortened telomeres. Overall, this manuscript represents a tour de force in quantitative genetic analysis of regulation of telomere length in a metazoan, identifying a NOP2 polymorphism that increases telomere length, whereas loss of function of NOP2 results in shortened telomeres.

Comments:

1. This manuscript is very clear.
2. Even if NOP2A and B paralogs redundantly affect ribosome biogenesis, it is interesting that the role in telomere regulation has been maintained by only one of the paralogs.
3. Could the phenotype imparted by SF-2 be due to higher *nop2a* expression? The text does not specifically say if or how expression differs in the SF-2 line (only that either expression differs or annotation is interesting). It might be possible to overexpress *nop2a* in order to achieve a similar telomere lengthening phenotype, in lieu of a rescue experiment. The authors mention doing RT-PCR to verify expression of the transgenes (line 191), but do not quantify expression levels.
4. Determining what sequence change in the SF-2 allele is causing the telomere phenotype would be interesting and useful in experiments to really drive home that the SF-2 *nop2a* allele is responsible for the phenotype. How quickly that can be done depends on how many candidate mutations there are. The ideal experiment would be targeted mutations using CRISPR, but the procedure is relatively new in Arabidopsis. If there are dozens of potential causative mutations then this might be beyond the scope of this manuscript.
5. Is there some way to show a map or spreadsheet of the unique sequence changes for SF-2 in and around the *nop2a* gene?
6. The authors mention expression patterns as part of the rubric for determining which candidate genes to investigate. However, they don't actually say what the difference was for the SF-2 allele of *nop2a*, if any.
7. It would obviously be great to work out the mechanism behind the telomere lengthening phenotype, but this would be open-ended and possibly require many experiments. This is probably more appropriate as the subject of a follow-up report.
8. Previous literature is well documented and the manuscript is easy to follow.
9. There are a few figure legends that could use some clarification. For the figure 1 legend, a few details of the boxplot could use explanation. What are the error bars? What are the black circles?

Alternatively, this could go in the methods section.

Minor comments:

1. I prefer to see the lower internal repeat bands, as appears in extended data figure 3C, just as a way to be assured of rough normalization of loading, but some people prefer not to include them as it necessitates much larger figures.
2. The ladder lane for figure 2A looks odd, like it was spliced onto the image. Sometimes lanes are not included in images, but might be better to line up the presented lanes with some space between them to be as transparent as possible.
3. Line 101+102. Were there any chromosome arms that were not affected? The way this is written, I cannot be certain. Were just these arms and the authors found that 5/5 were shortened, or did they test all arms and find 5/10 were shortened while the other 5 remained unchanged? I'm guessing it's the former. Please clarify.

Reviewer #1 (Remarks to the Author):

1. How many biological replicates were phenotyped for the founders, the MAGIC lines and the T-DNA lines? I assume there were biological replicates in order to estimate broad-sense heritability.

Yes, we did use biological replication (4-17 reps / genotype for the MAGIC parents, 2-8 for the T-DNA mutants, 1-2 for the MAGIC progeny). We have added statements to address these omissions in the main text [Page 2, line 67; Page 3, line 104; Page 5, line 231]. These data are also provided in Supplementary Data Tables 1, 2 and 5.

2. In the abstract it is stated that more than one QTL is mapped but I could find very little discussion of this point. If one performs a conditional analysis, either by removing those MAGIC lines (~1/19 or 4.5%) that carry Sf-2 at the major QTL, or by including the QTL haplotype as a covariate, do further QTL appear? In particular, can one rule out more than one causal variant contributing to the major QTL? Removing the major QTL would reduce the residual variance by about half and therefore could help clarify the significance of the QTLs elsewhere which just exceed the significance threshold.

This is a very good point, and one we should have considered previously. It is very true that controlling for the allelic variation of a large-effect QTL will increase the power to detect other minor QTL. We accomplished the suggested analyses by including, as an additive covariate, the Sf-2 genotype probabilities at the major QTL peak. In short, this analysis did improve the signal of a second peak on the proximate end of Chr5, which was not previously significant. It also improved the LOD score of the Chr 1 peak, which remained significant. Given the interesting additional results, and the improved signal at Chr 1, we have expanded our discussion of the ‘minor’ QTL in the main text [Page 2, lines 73-80] and now provide a new Supplementary Figure 1. This study also required a new set of analyses, which are detailed in the methods [Page 6, lines 238-245]

Above, you also bring up another point. ‘In particular, can one rule out more than one causal variant contributing to the major QTL?’ This is an important question, but also one that is not as easily addressed by additional analyses. Linkage mapping, even in highly recombinant and large populations, is limited by the degree of recombination and suffers from a lack of precision. It is also possible that multiple causal alleles exist under the QTL peak. These may all reside in a single gene, or in multiple genetically linked genes. Additional functional analyses are underway to test these hypotheses and we now discuss these alternatives in the main text [Page 3, lines 91-93].

3. How common is the Sf-2 haplotype in the 1001 Arabidopsis genome survey? Is there any evidence of geographical origin of the haplotype?

We have initiated this analysis, and considered including a map of *NOP2A* haplotype diversity. In short, we do find relatively interesting results that indicate that the Sf-2 haplotype sequence is quite distinct, yet also broadly geographically distributed. Overall, comparison of *NOP2A* sequences between 19 MAGIC founders indicates that there are

over 60 SNPs in the Sf-2 allele of *NOP2a* compared to the Col-0 reference and other founder lines, and that Bur-0 ecotype harbors the closest *NOP2A* sequence to Sf-2 allele. As Sf-2 and Bur-0 ecotypes are not from the same admixture cluster (Sf-2 is Iberian and Bur-0 is UK admixed), this finding argues against a specific geographical origin of the allele. However, we opted to exclude this analysis in the present manuscript for two reasons. First, experiments are underway to conduct more detailed molecular and functional analyses of *NOP2A* alleles (e.g. multigenerational complementation with multiple alleles and point mutants, etc.), so that we can make stronger inference between haplotype variation and the phenotype. Second, we felt that including an entirely new dataset, a new bioinformatic pipeline and the resulting paragraphs (or sections) of results is a bit too distinct from the primary goal of the present manuscript and more appropriate for a different manuscript that specifically addresses the genetic diversity of *NOP2A*.

4. More generally, what is the biological importance of this finding? Judging from figure 3, hets for the T-DNA insertions are recessive. However, the Sf-2 allele appears to increase telomere length, ie it is like a gain of function allele. Do the annotated Sf-2 sequence variations in *NOP2A* suggest a change to gene function (ie a changed protein sequence), or is the effect simply due to a change (presumably increase) in expression?

Our data indicate that while there are many amino acid changes present in 18 other MAGIC founder ecotypes compared to the reference Col-0, none of them are specific to Sf-2. As there are several Sf-2 specific nucleotide changes in the 3' and 5' regulatory regions of the gene, we suspected that *NOP2A* expression change in Sf-2 may be causal. Thus, we experimentally checked *NOP2A* gene expression in all 19 founder genotypes by qPCR (new Supplementary Figure 3B). Our experiments confirm the previously published data (ref. 42) that *NOP2A* expression in Sf-2 and Col-0 does differ by over 2-fold, but also indicate that gene expression alone is not sufficient to explain observed telomere length differences in the 19 founders. We now discuss these data in the main text (Page 3, lines 126-130). One conundrum is that there is a continuous distribution of telomere length among parents and MAGIC progeny. This is hard to explain by a binary *NOP2A* expression effect alone - presumably this is the result of other minor QTL or some interactions, along with some measurement error or environmental effects. Another explanation is that telomere length is a very unusual phenotype in that it is a direct molecular inheritance (a length of DNA) plus a very slow dynamic process that occurs over generations, as we are currently learning through the multigenerational analysis of *NOP2A* complementation lines. The MAGIC population represents only a handful of generations, so the setpoint of the parents may not be completely recovered in recombined progeny over the course of MAGIC breeding and in our experiments.

5. I assume the founder gene expression data used in the study is from ref 42 (Gan et al) - please clarify. What is the pattern of expression of NOP2A among the founders? There is also MAGIC line expression data available from Imprialou et al 2017 Genetics (available from GEO under series number GSE94107). These data would be useful to (a) confirm the founder expression patterns, (b) to see if the gene interaction module in Fig 4A is supported by MAGIC gene co-expression, (c) identify any other genes whose expression is correlated with NOP2A.

You are correct, the founder gene expression data is from ref 42. This is now more explicitly laid out [Page 3, line 95].

We had not previously considered expression data from the MAGIC lines; however, these data certainly do offer the potential for more detailed empirical tests of the hypotheses laid out in the interaction modules and to potentially dissect the patterns of expression underlying the primary QTL. However, as we discuss in our response (to point 4) above, we believe that transcript abundance alone is not sufficient to explain observed telomere length differences in the 19 founders, so we do not necessarily expect a large degree of expression variation as a driver of the QTL. However, to address the points brought up above, we conducted a new set of analyses on data from Imprialou et al. (2017). In short, we find that for expression co-variation largely re-capitulates founder expression between the Sf-2 / non-Sf-2 haplotypes at the *NOP2A* QTL. We now present this result in the main text [Page 3, lines 118-125] and Figure S3A.

6. Please clarify if there is a distinction between telomere set point and telomere length.

Telomere length is a general term referring to the average length of telomeric DNA tracts. Telomere length is highly dynamic and, in humans, shortens over time in most somatic cells. **Telomere length set point** refers to a species- or genotype-specific telomere length homeostasis achieved through the balance in the forces that extend the telomere tract (e.g. the telomerase enzyme and recombination) and forces that shorten it (e.g. end replication problem, nucleolytic attack, deletional recombination). Thus, telomere length set point can be viewed as a more specific term reflecting telomere homeostasis that is achieved through successive organismal or cellular generations. We now clarify this term in the main text [Page 2, lines 47-48].

Reviewer #2 (Remarks to the Author):

Abdulkina et al. describe the role of NOP2A, an RNA methyltransferase involved in ribosome biogenesis, in telomere length set point establishment. For this, they exploit a genetic approach aiming at identifying the causal factor for natural variation in telomere length in *Arabidopsis thaliana*. Although they convincingly demonstrate that NOP2A, and other players in ribosome biogenesis, is involved in regulating telomere length set point, I am less convinced that allelic variation for this gene explains the natural variation observed.

For instance, the broad sense heritability among parental genotypes is very high and the range of variation is quite wide, with evenly distributed phenotypic values along the spectrum (Fig. 1). Yet,

a QTL analysis only detects a single relevant QTL, which only seems to explain the extreme phenotype of accession Sf-2 (Fig. 2). So, given the high heritability, what does explain the large variation in telomere length in the other accessions and why could this not be mapped?

This is a good point, and one that we did not previously discuss at enough length in the manuscript. The MAGIC individuals with very long telomeres almost all have the Sf-2 allele at the main QTL. Hence the large amount of phenotypic variance explained by this QTL. However, it certainly begs the question of what is controlling the remainder of the heritable genetic variation. The first answer is that there was another minor QTL on Chr 1, which we brushed over before, and now discuss directly [Page 2, lines 73-80]. Second, as discussed in response to Reviewer 1's point 2 above, we have now performed an additional analysis by controlling for the allelic variation of a large-effect QTL on Chr 5, increasing the power to detect other minor QTL. This analysis not only improved the signal of a minor QTL on Chr 1, but also detected an additional second peak on Chr5, which was not previously significant. We now provide a new Supplementary Figure 1 and discuss these data in the text [Page 2, lines 73-80]. Finally, there are undoubtedly many small-effect loci that combine to explain a large amount of remaining phenotypic variance, but do not individually have enough effect to be significant at the genome-wise error rate correction that we conduct.

Is it possible that the true causal gene was not selected for KO screening? The phenotype suggests a gain of function in Sf-2 compared to all other accessions. This is hard to proof with a KO or sequence analysis. If the causal gene is a repressor, however, a KO would mimic the Sf-2 phenotype and loss of function mutations are much better predictable and more common. I have no doubt that NOP2A plays a role in telomere length but question whether this gene is linked to the observed variation.

Yes, it is possible that other genes are involved in the QTL effect observed. We have not screened all possible candidate genes in the QTL interval and it is not uncommon for major effects to be the result of multiple causal mutations. However, the list of candidate genes is long and it has been an ambitious effort to screen the 20 candidates presented in the manuscript. Here, we focus on the discovery of NOP2A effects and feel this is appropriate given the novelty of the discovery. We believe the fine mapping and study of other QTL or candidates is beyond the scope of the current study.

Furthermore, they test a KO of NOP2A in the Col background, which is one of the accessions with the shortest telomeres already, to show that a null-mutant results in even shorter telomeres, while the Sf-2 allele should result in longer telomeres. Perhaps a Crispr/Cas KO in the Sf-2 background provides more insight on the causality of the NOP2A locus. It would also be good to know the sequence variation between Col and Sf-2 and its anticipated effect.

As stated above, the QTL doesn't have to resolve to a single SNP per se, and could be more subtle phenomena, including potential interactions between SNP sites or other loci. While it is impossible to know the causal polymorphism(s) without additional fine-mapping and functional tests (which have been initiated, including the Crispr/Cas KO in the Sf-2 background), alternative scenarios can also include epistasis with other genes,

posttranslational modifications of *NOP2A*, tissue or development-specific expression or splice variants, etc. Detailed functional tests are underway but are outside of the main scope of this manuscript.

A complementation test might also provide evidence for allelic variation at the *NOP2A* locus explaining the variation between Col and Sf-2. However, this test seems to do the opposite. Allelic variation in the Fs-2 gene does not lead to a difference in phenotype and complementation does not result in Fs-2 like trait values (carefully hidden in the extended data figure 2), making it unlikely that *NOP2A* is the causal gene underlying the QTL.

We clearly acknowledge in the manuscript that the transgenic complementation tests do not unambiguously confirm that natural alleles of *NOP2a* underlie the observed QTL [Page 3, lines 134-136]. Rather, the experiment shows that both Col and Sf-2 alleles are functional and result in similar complementation of the KO in the first (T1) generation. There are several possible explanations for this result. It is possible that we have the wrong gene, although we think this is highly unlikely given the uniqueness of telomere set point mutations. We think it is more likely that aspects of the transgenic experiment limit our inference. It is not uncommon for transgenic complementation of natural alleles to fail due to positional effects, because the causal polymorphisms are regulatory and require specific expression characteristics, or because of complex dominance or epistasis. We think a more likely explanation in this case is that telomere set point is a dynamic phenotype that requires multiple generations to accrue (ref 6). We now mention this as a possibility [Page 3, line 137-138]. Thus, in this manuscript we carefully avoid claiming that *NOP2A* is the causal gene underlying the QTL – only that based on all our assays and supporting data it is the most likely candidate.

So, while a carefully conducted genetic screen identifies a strong QTL explaining variation in telomere length, it turns out that this QTL only explains (perhaps only partly) the extreme phenotype of accession Sf-2. They further provide compelling evidence that *NOP2A* and related genes are involved in the regulation of telomere length. However, they fail to prove that *NOP2A* is actually the gene underlying the detected QTL, nor do they provide additional loci that might explain the highly heritable variation between the other accessions.

We agree that we had previously not spent enough time discussing the quantitative genetics of telomere length variation aside from *NOP2A*. We have now substantially expanded corresponding sections detailing our analysis of the two minor QTL that help explain the highly heritable variation, and the effect of polygenic (many small QTL) inheritance in the main text (Page 2, lines 73-80; Page 2, lines 91-93). We are also careful in our inference about the role of *NOP2A* in underlying the major effect QTL.

Finally, the authors repeatedly hint at the evolutionary conserved role of *NOP2* and speculate that this gene might also determine telomere length in humans and other species. I would argue that when this aspect is emphasized so strongly some empirical evidence supporting this theory could be expected.

We agree with the Reviewer that we did not discuss this information well enough in the previous version of the text. The evolutionarily conserved role of NOP2 proteins in ribosome biogenesis has been well-documented (refs 22-26). Specifically, Bourgeois et al. (ref 22) verified this by performing successful functional complementation of Nop2-deficient yeasts by human NOP2 orthologue and characterizing conserved domains necessary for correct protein localization and its cellular function in m5C modification of rRNA and pre-rRNA processing. We now more specifically emphasize these earlier findings in the main text (Page 3, lines 108-111).

Given all the known conserved functions of NOP2 throughout eukaryotic evolution (rRNA modification and ribosome assembly, cell cycle and cell proliferation, refs 22-23, 25-26), it does not seem unreasonable to speculate that the telomere function of NOP2 might also be conserved. This idea is further supported by a recent discovery of a physical association between human NOP2 and the catalytic telomerase subunit hTERT that was shown to stimulate a non-telomere function of telomerase, transcription of G1 phase cyclin D1 (ref 38, also discussed in the main text, Page 4, lines 188-189). Nevertheless, we agree with the Reviewer that the putative telomere role of human NOP2 needs to be tested directly. We have now modified the last paragraph of the manuscript to more specifically emphasize this point [Page 4-5, lines 189-191].

Reviewer #3 (Remarks to the Author):

1. This manuscript is very clear.
2. Even if NOP2A and B paralogs redundantly affect ribosome biogenesis, it is interesting that the role in telomere regulation has been maintained by only one of the paralogs.
3. Could the phenotype imparted by Sf-2 be due to higher *nop2a* expression? The text does not specifically say if or how expression differs in the Sf-2 line (only that either expression differs or annotation is interesting). It might be possible to overexpress *nop2a* in order to achieve a similar telomere lengthening phenotype, in lieu of a rescue experiment. The authors mention doing RT-PCR to verify expression of the transgenes (line 191), but do not quantify expression levels.

As previously reported (42) and confirmed by our qPCR analysis (new Supplementary Figure 3B), *NOP2A* expression in Sf-2 is indeed higher than in Col-0, but our data also indicate that gene expression alone cannot explain observed telomere length differences in the 19 founders. Rather, our ongoing experiments suggest a constant but very slow rate of telomere length increase in several generations of all *nop2a-2* knockout lines complemented with Sf-2 and Col-0 *NOP2A* alleles. As such, we favor the hypothesis that the effect of Sf-2 allele can only be detected after analyzing multiple consecutive generations of the complementation lines. These multigenerational experiments are currently underway.

4. Determining what sequence change in the SF-2 allele is causing the telomere phenotype would be interesting and useful in experiments to really drive home that the SF-2 *nop2a* allele is responsible for the phenotype. How quickly that can be done depends on how many candidate mutations there are. The ideal experiment would be targeted mutations using CRISPR, but the procedure is relatively new in Arabidopsis. If there are dozens of potential causative mutations then this might be beyond the scope of this manuscript.

See below

5. Is there some way to show a map or spreadsheet of the unique sequence changes for SF-2 in and around the *nop2a* gene?

Response to points 4 and 5:

As now discussed in [Page 3, lines 118-120], over 60 SNPs are present in the 18 MAGIC founder genomes compared to the Col-0 reference throughout the *NOP2a* gene (see also response to point 3 of Reviewer 1), though Sf-2 specific SNPs are localized only in the 3' and 5' regulatory regions, which are more difficult to dissect. We have indeed undertaken CRISPR experiments attempting to knock out *NOP2A* gene in the Sf-2 background, but have so far failed to generate any mutants with this technology. Further dissecting the exact nature of the causal *NOP2A* polymorphism will continue to be the focus of our future investigations.

6. The authors mention expression patterns as part of the rubric for determining which candidate genes to investigate. However, they don't actually say what the difference was for the SF-2 allele of *nop2a*, if any.

Good question and one also raised by reviewers 1 and 2 (see our response above). We have added a new Supplementary Figure 3, Supplementary Table 6 and discussion in the main text to this end (Page 3, lines 122-130).

7. It would obviously be great to work out the mechanism behind the telomere lengthening phenotype, but this would be open-ended and possibly require many experiments. This is probably more appropriate as the subject of a follow-up report.

Correct – we are currently undertaking several addition analyses, but the results are still months/years out.

8. Previous literature is well documented, and the manuscript is easy to follow.

9. There are a few figure legends that could use some clarification. For the figure 1 legend, a few details of the boxplot could use explanation. What are the error bars? What are the black circles? Alternatively, this could go in the methods section.

We have added clarification in the Figure 1 legend (in red). (Page 10, lines 412-414)

Minor comments:

1. I prefer to see the lower internal repeat bands, as appears in extended data figure 3C, just as a way to be assured of rough normalization of loading, but some people prefer not to include them as it necessitates much larger figures.

As figures in the main body of the manuscript are already complex with multiple panels, the bottom of each TRF gel was cropped to meet the figure guidelines. However, as suggested by the Reviewer, we have modified the corresponding Supplementary figure (now number 5) to include the lower internal repeat bands for all panels, as a way to demonstrate rough normalization of loading (Page 18).

2. The ladder lane for figure 2A looks odd, like it was spliced onto the image. Sometimes lanes are not included in images, but might be better to line up the presented lanes with some space between them to be as transparent as possible.

We thank the Reviewer for noticing this point; the image was adjusted according to the Reviewer suggestion.

3. Line 101+102. Were there any chromosome arms that were not affected? The way this is written, I cannot be certain. Were just these arms and the authors found that 5/5 were shortened, or did they test all arms and find 5/10 were shortened while the other 5 remained unchanged? I'm guessing it's the former. Please clarify.

In Arabidopsis, not all chromosome arms are amenable to measuring telomere length by PETRA as not all of them possess unique subtelomeric DNA sequences. For example, telomeric DNA on two chromosome arms (2L and 4L) is adjacent to rRNA clusters located immediately proximal to the start of each telomere repeat sequence, and 1R and 4R subtelomeres share an extensive region of homology (ref 41). Thus, we only measured telomere length on 5 individual chromosome arms: 1L, 3L, 4R, 5L and 5R. Telomeres of all tested chromosome arms (5/5) were shortened in all *nop2a* mutants. We have added these clarifications to the main text (Page 3, lines 114-116)

REVIEWERS' COMMENTS:

Reviewer #1 (Remarks to the Author):

The authors have dealt with the points I raised in my review satisfactorily.

I raise two minor points which I don't think are essential for the authors to deal with, but I would encourage them to do so as they are both easy and would improve the manuscript.

1. My original query regarding the presence of multiple causal variants at the major QTL was perhaps unclear - what I meant was for them to determine if there was any significant residual QTL at that locus after regressing out the peak signal. This should be clear from the analysis they have now done, so it should be trivial to answer this point.

2. Although I don't absolutely insist on it, I think it would be very worthwhile checking if there is a significant correlation in the MAGIC lines between NOPA2 expression and TRP levels, analogously to the plot in extended data figure 3B for the MAGIC founders.

Otherwise I think it is a nice study.

Richard Mott

Reviewer #2 (Remarks to the Author):

Unfortunately, I adhere to my earlier point of view that there is little wrong with the evidence presented that NOP2A is involved in regulating telomere set point but that the authors fail to prove that this gene is the underlying causal factor explaining the observed natural variation. For instance, they state that - 'the MAGIC individuals with very long telomeres almost all have the Sf-2 allele at the main QTL.' - but that is exactly my point; the Sf2 allele is the only cause of substantial variation. While this may explain some of the variation in the MAGIC population it cannot explain the difference between the parental accessions. So, my earlier criticism was not addressed, except that they now describe two additional, minor, QTL, detected after co-factor analysis, that might explain the extra variation. These, however, were of modest effect and certainly cannot explain all the variation.

Furthermore, simply stating that addressing my comments would be beyond the scope of the current study is the same as ignoring the issues raised, which is inappropriate.

In fact, they changed very little in the revision to alleviate my earlier concerns. Moreover, the additional evidence presented, such as the lack of functional SNPs and the expression data, rather hint at the opposite: Variation in NOP2A cannot explain the observed phenotypic variation, as should also be concluded from the KO studies. I agree that it can be difficult to link genotypic variation to phenotypic variation by KO and expression studies but in this manuscript there is not a shred of evidence to suggest that NOP2A is the causal gene. Rather the opposite, all evidence points to rejecting NOP2A as a candidate. As the paper is presented as a mapping study this is a major flaw.

I am not sure how to address this or whether it is even relevant to do so. It is clear that the NOP2A gene was identified by co-location of a major QTL and its role in telomere length control is also evident. However, validation of NOP2A as the causal gene underlying the major QTL could not be provided and experimental results should not be presented as evidence for this if it clearly points in the other direction. Perhaps a simple statement at the end stating that the validation was not successful, despite several attempts, but that NOP2A is undoubtedly involved in telomere lengthening would be sufficient.

Reviewer #3 (Remarks to the Author):

The authors have provided a strong set of responses to the Reviewers' comments. The authors demonstrate that three loci repress telomere length in wild Arabidopsis strains. The major effect locus occurs at the distal end of chromosome 5 and includes several hundred genes. The authors studied gene expression and identity, and then used gene knockouts to show that the NOP2a locus promotes telomere elongation. They acknowledge that it is possible that other genes may contribute to telomere homeostasis in this interval, but they show strong supporting data that distinct ribosome biogenesis proteins promote telomere elongation. This is an elegant and carefully conducted study that is supported by a recent publication that suggests that NOP2 may interact with telomerase.

I suggest that the authors create supplemental information that shows the polymorphisms from the Sf-2 background and their locations within the 5' and 3' regions of NOP2, as they have this data but do not explicitly show it. This might help to show which SNPs affect 5' or 3' UTRs of NOP2a in a manner that could affect RNA stability or translation, and which might be transcriptional regulators. In addition, I would suggest a supplemental file that contains all the candidate genes within the Chr 5 locus and their predicted protein products. Genome wide analysis of effects of gene knockouts on telomere length in *S. cerevisiae* has revealed many genes regulate telomere length, even though they might not be 'expected' to regulate telomere length based on the proteins they encode.

Reviewer #1 (Remarks to the Author):

The authors have dealt with the points I raised in my review satisfactorily.

I raise two minor points which I don't think are essential for the authors to deal with, but I would encourage them to do so as they are both easy and would improve the manuscript.

1. My original query regarding the presence of multiple causal variants at the major QTL was perhaps unclear - what I meant was for them to determine if there was any significant residual QTL at that locus after regressing out the peak signal. This should be clear from the analysis they have now done, so it should be trivial to answer this point.

A: When we regress out the allelic effects of the primary Chr5 QTL, we do see a small remnant peak with an effect unrelated to the Sf-2 allele. Therefore, while it is possible (even likely) that other loci contribute to the morphology of the observed large QTL, it is clear that allelic effects of other nearby loci do not drive the significance of the peak. This observation did not require any additional analysis, but we have now added this clarification as Supplementary Figure 1B and additional text (page 4, lines 116-117).

2. Although I don't absolutely insist on it, I think it would be very worthwhile checking if there is a significant correlation in the MAGIC lines between NOPA2 expression and TRP levels, analogously to the plot in extended data figure 3B for the MAGIC founders.

A: We found little evidence of a correlation between *NOP2A* expression and mean TRF in the MAGIC population. However, there is also very little replication of the Sf-2 allele for *NOP2A* (just six lines), so it is not clear whether the lack of signal is due to no correlation in the population, or just no correlation in non-Sf2 *NOP2A* alleles. This clarification has now been added as Supplementary Figure 3C and additional text (page 5, lines 173-175).

Otherwise I think it is a nice study.

Richard Mott

Reviewer #2 (Remarks to the Author):

Unfortunately, I adhere to my earlier point of view that there is little wrong with the evidence presented that *NOP2A* is involved in regulating telomere set point but that the authors fail to prove that this gene is the underlying causal factor explaining the observed natural variation. For instance, they state that - 'the MAGIC individuals with very long telomeres almost all have the Sf-2 allele at the main QTL.' - but that is exactly my point; the Sf2 allele is the only cause of substantial variation. While this may explain some of the variation in the MAGIC population it cannot explain the difference between the parental accessions. So, my earlier criticism was not addressed, except that they now describe two additional, minor, QTL, detected after co-factor analysis, that might explain the extra variation. These, however, were of modest effect and certainly cannot explain all the variation.

Furthermore, simply stating that addressing my comments would be beyond the scope of the current study is the same as ignoring the issues raised, which is inappropriate.

In fact, they changed very little in the revision to alleviate my earlier concerns. Moreover, the additional

evidence presented, such as the lack of functional SNPs and the expression data, rather hint at the opposite: Variation in NOP2A cannot explain the observed phenotypic variation, as should also be concluded from the KO studies. I agree that it can be difficult to link genotypic variation to phenotypic variation by KO and expression studies but in this manuscript there is not a shred of evidence to suggest that NOP2A is the causal gene. Rather the opposite, all evidence points to rejecting NOP2A as a candidate. As the paper is presented as a mapping study this is a major flaw.

I am not sure how to address this or whether it is even relevant to do so. It is clear that the NOP2A gene was identified by co-location of a major QTL and its role in telomere length control is also evident. However, validation of NOP2A as the causal gene underlying the major QTL could not be provided and experimental results should not be presented as evidence for this if it clearly points in the other direction. Perhaps a simple statement at the end stating that the validation was not successful, despite several attempts, but that NOP2A is undoubtedly involved in telomere lengthening would be sufficient.

A: As stated in the previous revision, we agree with the Reviewer that we do not currently have unambiguous evidence that the identified gene is the only underlying causal factor explaining the observed natural variation. It is possible that other genes may also contribute to telomere homeostasis in this interval or underlie the QTL effect observed.

The Reviewer suggests: “perhaps a simple statement at the end stating that the validation was not successful, despite several attempts, but that NOP2A is undoubtedly involved in telomere lengthening, would be sufficient”. To reconcile our manuscript with this suggestion, we indeed provide an extended version of such statement on page 5 (lines 181-189).

Reviewer #3 (Remarks to the Author):

The authors have provided a strong set of responses to the Reviewers' comments. The authors demonstrate that three loci repress telomere length in wild Arabidopsis strains. The major effect locus occurs at the distal end of chromosome 5 and includes several hundred genes. The authors studied gene expression and identity, and then used gene knockouts to show that the NOP2a locus promotes telomere elongation. They acknowledge that it is possible that other genes may contribute to telomere homeostasis in this interval, but they show strong supporting data that distinct ribosome biogenesis proteins promote telomere elongation. This is an elegant and carefully conducted study that is supported by a recent publication that suggests that NOP2 may interact with telomerase.

I suggest that the authors create supplemental information that shows the polymorphisms from the Sf-2 background and their locations within the 5' and 3' regions of NOP2, as they have this data but do not explicitly show it. This might help to show which SNPs affect 5' or 3' UTRs of NOP2a in a manner that could affect RNA stability or translation, and which might be transcriptional regulators.

A: We now provide Supplementary Data 6 that shows *NOP2A* nucleotide polymorphism in Col-0 and Sf-2 parents.

In addition, I would suggest a supplemental file that contains all the candidate genes within the Chr 5 locus and their predicted protein products. Genome wide analysis of effects of gene knockouts on telomere length in *S. cerevisiae* has revealed many genes regulate telomere length, even though they might not be 'expected' to regulate telomere length based on the proteins they encode.

A: This information is provided in the Supplemental Data 3 file, which contains all the candidate genes within the Chr 5 locus and their predicted protein products.